# Intersectoral Actions for the Promotion and Prevention of Obesity, Diabetes and Hypertension in Brazilian Cities: A Systematic Review and Meta-Analysis

**DOI:** 10.3390/ijerph192013059

**Published:** 2022-10-11

**Authors:** Stephen Kunihiro, Juliana Ribeiro da Silva Vernasque, Celso da Silva, Marcela Facina dos Santos, Camila Pires Cremasco, Luís Roberto Almeida Gabriel Filho

**Affiliations:** 1School of Sciences and Engineering, São Paulo State University (UNESP), Tupã 17602-673, Brazil; 2Faculty of Medical and Biological Sciences, São Paulo State University (Unesp), Botucatu 18618-687, Brazil; 3Marília Medical School, FAMEMA, Marília 17519-030, Brazil; 4Maurício de Nassau College—Dean’s Office, Maurício de Nassau University Center (UNINASSAU), Recife 52011-220, Brazil; 5The Regional Council of Administration—CRA/SP, São Paulo 01427-001, Brazil

**Keywords:** intersectoral actions, diabetes mellitus, hypertension, obesity, prevention of diseases, non-communicable chronic diseases, NCDs

## Abstract

This study showed the effectiveness of biomedical interventions in obesity, diabetes and hypertension (NCDs), but innovative and intersectoral elements in the fight against obesity, type 2 diabetes and hypertension were rare. Background: Is it possible to find effective and innovative actions to promote health and prevent NCDs in Brazilian municipalities? Can they be replicated? Objective: Our objectives were to identify innovative and effective intersectoral actions for promoting and preventing NCDs in Brazilian municipalities. Methods: This is a systematic review in an exploratory theoretical essay with a qualitative and quantitative approach. It is descriptive and analytical in terms of reporting findings and results. Inclusion and exclusion criteria favored health promotion work. Bias risk assessments was performed using the Cochrane GRADE and bias risk, with meta-analyses using RevMan and Iramuteq. Results: Meta-analysis of biometric markers resulted in −4.46 [95% IC; −5.42, −3.49], *p* = 0.00001, indicating a reduction in NCD risk rates. The textual meta-analysis revealed P(r) ≈ 83% (Reinert), meaning low connectivity between the ‘halos’. Conclusions: There is evidence of the effectiveness in interventions, but innovative and intersectoral elements to combat and prevent NCDs were barely seen. While evidence of intervention effectiveness was observed, innovative and intersectoral elements to combat and prevent NCDs were barely noticed.

## 1. Introduction

In the global context, even with advances in the SDG (Sustainable Development Goals) 2030 agenda, problems related to *“undernutrition, overweight and obesity in children, continue to be a major concern. In addition, maternal anemia and obesity among adults remains alarming.”* [1] (p. 8).

Recent evidence indicates that there has been a surge from 112 million to almost 3.1 billion people who are unable to access healthy food. It was also found that overweight or obese children suffer immediate and long-term damage, with a higher risk of NCDs. The prevalence of obesity in children under five years old in the last 10 years has increased from 5.4 percent (33.3 million) in 2000 to 5.7 percent (38.9 million) in 2020 [1].

The diagnostic criteria for metabolic syndrome endorsed by the World Health Organization, Third Report of the National Cholesterol Education Program Expert Panel on Detection, Evaluation and Treatment of High Blood Cholesterol in Adults and the International Diabetes Federation, is based on WHR—waist/hip ratio, BMI—body mass index, either high triglycerides or low HD, HDLs—high density lipoproteins, SBP—systolic blood pressure, DBP—diastolic blood pressure and SAH—systemic arterial hypertension [2]. Obesity, type 2 diabetes and insulin resistance are identified as the main etiopathogenic factors that generate glucose and lipid metabolic disorders with increased cardiovascular risk, which is the main cause of mortality, especially in emerging countries such as Brazil [3].

Regarding the SINDEMIA and World Burden Morbidity and Mortality (GBD), there are notes related to great physical and emotional pain [4,5], progressing to degenerative diseases such as cancer, up to premature death [6], with higher prevalence in men in groups with lower education and in women with low income [7,8,9].

In the sphere of molecular genomic studies, the *PPP1R3B* gene is associated with the body’s ability to accumulate carbohydrates and lipids and, consequently, a higher predisposition to develop obesity, which is related to metabolic syndromes such as type 2 diabetes [10]. Obesity and diabetes genes are more present in some individuals than in others due to the genetic load of ancestry, which forms an even more complex and comprehensive picture when it is combined with epigenetics, which is itself entangled in the most varied intersectoral multi- and interdisciplinary webs of triggers, such as chronic stress and metabolic syndrome, nutritional and neurological dysfunction, socio-emotional breakdown, physical inactivity, excessive exposure to smartphone, tablet or computer screens and non-stop browsing on social media associated with the abusive consumption of high-caloric and low-nutrition food. All these elements contribute to aggravate NCDs [11,12].

Certain studies seek greater effectiveness in combating NCDs in school environments by providing educational activities, [7,9], although children and adolescents in place of health, seek pleasure, by satisfying themselves with low-nutrition and high-calorie foods [13,14]. Acquired habits are the results of some factors such as insufficiency of affordable food products, poor eating habits and orientation towards healthy eating as well as the insertion in an overall unhealthy environment [15]. Some studies have proven the correlation between overweight children and losses in cognitive abilities and cognitive skills loss, leading to learning deficits, and subsequent school dropout [16]. Later on in life, these children are hampered by reduced opportunities both for good jobs and better wages, impacting later on in the major rise of NCDs [17] and reduced life expectancy [18,19].

The budget applied to reduce the consequences of NCDs has been equivalent to 70% of all hospitalization costs from ICU, surgery and medication, among others. In order to reduce premature death, approximately US$ 3.45 billion/year is spent due to faulty management and the aggravation of NCDs [20]. The ineffective efforts to reduce the causes, proportional to the global scope, of the global obesity syndemic/WHO 2018/URBAN 95 [21] led the UN to challenge nations with the Sustainable Development Goal—SDG 20/30. Brazil in 2017, assumed the following objectives:Goal 2.1—United Nations/Brazil: ensure access for all people, in particular for vulnerable and poor people, including children, to safe, nutritious and sufficient food throughout the year;Goal Brazil 2.2—reduce and eradicate forms of malnutrition related to overweight or obesity.Goal 3.4—United Nations/Brazil: reduce premature mortality from non-communicable diseases by one third through prevention and treatment—Indicator 3.4.1: mortality rate from diabetes mellitus;Goal 4.2—United Nations: ensure that all girls and boys have access to quality early childhood development, care and pre-school education so that they are ready for primary education—Brazil 4.2.1: Proportion of children under 5 years of age who have adequate health, learning and psychosocial well-being [22].

According to the WHO, *“investing, every year, US$0.84 per person to prevent disease in early stage could yield around US$230 billion in economic gains, seven million lives saved, 10 million heart attacks and strokes avoided”* [23]. *“With the right strategic investments, countries that carry a significant amount of NCDs burden can change their disease trajectory and provide significant health and economic gain for their citizens”* [23].

Based on these arguments, the theme of this research was to present innovative actions to promote health and prevent obesity, diabetes and hypertension in Brazilian cities, whether such actions are, in fact, effective, and whether they can be replicated.

Given the relevance of the theme, this research is justified. The research objective was defined: to identify innovative and effective intersectoral actions in Brazilian municipalities for the prevention of obesity, hypertension and diabetes through a systematic review (SR) and meta-analysis.

The hypotheses of this study are:

$H_{0}$: Interventions for prevention and treatment of obesity, diabetes and hypertension are effective.$\;H_{1}$: Interventions to prevent and treat obesity, diabetes and hypertension are not effective;$H_{0}$: There are innovative and intersectoral elements in the actions to combat and prevent NCDs. $H_{1}$: There is an absence of innovative and intersectoral elements in actions to combat and prevent NCDs;$H_{0}$: By adapting to local socioeconomic and cultural conditions, effective innovative and intersectoral actions are replicable. $H_{1}$: Even with all necessary adjustments, effective innovative and intersectoral actions are not replicable.

For the sake of reaching the objective, a meta-analysis of the data presented in the studies and the textual corpus of the studies was defined, with indicators and results based on retrieved data.

## 2. Materials and Methods

### 2.1. Search Strategy

The methodology applied in the study consisted of a theoretical exploratory essay with a qualitative and quantitative approach through systematic review (SR), so as to test the hypothesis within protocols supported by evidence-based medicine [24]. The methodological documentation was descriptive and analytical regarding the reports of findings and of results [25] through meta-analysis of data and textual studies [26].

### 2.2. Selection, Inclusion and Exclusion of Studies

The systematic review complies with meta-analysis guidelines: Preferred Reporting Items for Systematic Reviews and Meta-Analysis (PRISMA 2020), registered in the PROSPERO database under ID: CRD42022319835 in addition to an investigative protocol and data management plan in DMPTool database with DOI: 10.48321/D1K31N. Eligibility criteria were established in terms of inclusion and exclusion variables, data extraction and synthesis procedures, as well as meta-analysis and quality assessment [27,28].

The search strategies included surveys in the databases of the following repositories: Lilacs, PubMed, Scopus, Web of Science, Medline, and EMBASE.

All examination procedures were performed in pairs by two researchers and an additional third party was included wherever any points of divergence and disagreement were raised whereby the most reasonable position prevailed in case of irreconcilable opinions. The inclusion and exclusion criteria were established based on the research objectives and were enforced by the researchers, assisted by StArt—State of the Art Through Systematic Review software [29].

### 2.3. Data Extraction and Synthesis

Data extraction was adapted from the Cochrane Database of Systematic Reviews tool: Working Group on Classification of Recommendations, Evaluation, Development and Evaluation—GRADE® [30,31,32].

A meta-analysis was performed by Cochrane’s RevMan® 5.4.1 software in order to obtain the combined effect, the statistical synthesis of the effect added to effectiveness of the interventions [33,34], with textual meta-analysis using Iramuteq® 0.7 alpha 2 software (LERASS, Toulouse, France) [35].

### 2.4. Quality Assessment

Quality assessment comprised each step of the selection, extraction and summarization processes with assessment filters. StArt software generated a score adjusted to the key incidence searching terms in the studies. In extraction, through the GRADE model, typological questions and factors that could decrease and increase the quality of evidence were evaluated [30,31,32], generating a quality score on a scale from 1 to 9, considering: 1 to 3—limited importance (very low); 4 to 6—important (high); and 7 to 9—critical (very high) [30,31]. It was established that articles reaching a score equal to/greater than 4 would be included in the extraction phase.

In the summary for synthesis, risk of bias analysis was carried out, comprising the dimensions of the Cochrane Risk of Bias Protocol—ROB2 in its ninth edition [36,37,38]. Once aware of elements with certain amounts of risk of bias or uncertainty, the results were submitted to a normality test with Minitab Statistical Software 19 to evaluate the measured effects [39].

The synthesis was performed by meta-analysis of intervention data using RevMan 5.4.1 software [40], and textual meta-analysis using Iramuteq 0.7 alpha 2 software [41]. To minimize possible risks of bias, the procedures were paired and revised. Finally, contour referencing was conducted by Zotero software.

## 3. Results

### 3.1. Selection of Studies

The research results reported in this section were achieved by applying a structured and documented protocol, with the steps of the process represented by the systematic review flow diagram represented in Figure 1.

The flow diagram presents 748 studies identified in the search procedures; after the selection and extraction procedures, 27 qualified studies were extracted for summarization and meta-analysis based on the protocol of this research.

### 3.2. Characteristics of the Studies

Identification of the studies’ characteristics was carried out after analyzing the different contexts and effectiveness of intersectoral actions for prevention of obesity, hypertension and diabetes investigated in terms of innovation, as shown in the Table 1.

In Table 1, there is a summary of the 27 articles that were exhaustively analyzed in different contexts, receiving scores according to different levels of importance considering the result of the analysis to answer whether the studies were effective, innovative, intersectoral and replicable. 

Regarding the effectiveness of the preventive process (often educational) verified after a long period of follow-up post-intervention, Ferreira and Vasconcellos, despite their small sample size and short period of intervention, demonstrated effectiveness in reducing cardiovascular risk factors using rigorous clinical protocols and precision medicine with specific biomarkers, increasing the availability of resources and necessary skills for accurate implementation and management [54,66]. Post-intervention is the key factor for a well-executed educational process with continuous and ongoing adjustment for the effectiveness of the NCD prevention program [52]. Leme, over the course of a fairly brief period of 6 months carried out an educational intervention to combat the harmful habits that lead to obesity. Nonetheless they did not include biomedical or anthropometric results (outcomes) [56]. Santos presented an intervention with follow-up of 15 years, where a biomedical-based program for metabolic issues assisted by non-biomedical actions such as nutrition and physical activity has proven to be effective through multiple biomarkers [63].

#### 3.2.1. Health Innovation with a Focus on Biomedical Technology 

Clinical efficacy-focused studies have been using high medical precision innovative elements, making it quite costly to replicate into a large portion of the NCD population. A precise multicomponent physical activity intervention by Coelho found a decrease in blood pressure for elderly population hypertensive patients, in a short 6-month period of time [48], with the following results:(a)Effectiveness: It was shown to be effective by the biomedical approach with high bio-chemical mediators;(b)Innovation: Intervention involved high technological commitment, with sophisticated mechanisms of motivation for engagement and use of biomechanical components replacing ergometric equipment;(c)Limitations: Considering feasibility and replicability, as well as cost-effectiveness and cost-benefit, the study was shown to be very effective, focusing on precision. Nevertheless, it was found very costly, making it inaccessible to the vast majority of the NCD population. The focus on technological effectiveness, excludes several aspects of society such as socioeconomic and contextual anthropological approaches. The absence of these elements denotes the lack of a broader and intersectoral view as well as the social determinants of health [48].

#### 3.2.2. Health Innovation with a Focus on Social Technology

Two articles brought some sociological elements into analysis, specifically netnography and the use of the WhatsApp application. Both of them showed how, in a short period of time, social media could present some relevant sociological findings related to health. These articles presented some evidence about these technologies regarding increased adherence and engagement for self-care and continuous ongoing monitoring of NCDs.

Effectiveness and innovation: The continuous use of medications for the treatment of hypertension and diabetes using WhatsApp resulted in a 15% increase in adherence to medication, as evidenced by the magnitude of the effect measured verified by relative risk [65]. The netnographic study by Fernandes spontaneously generated a large number of posts with relevant information about the disease and treatment, where participants shared their own experiences, providing social support, interactions and subsequent reactions, such as comments, likes and reposts [52].

Limitations and replicability: One of the limiting factors in netnography, especially in social networks such as WhatsApp, is the fact that it produces non-normative information and much of it is inconsistent, with no relation to scientific evidence. Motivation based on people’s experiences and feelings, cause an apparent adherence to NCD health care. However, over a period of time, they tend to return to the original state, the old practices, which is known as the Hawthorne effect, an immediate result of movement towards mass communication and low criticality. As a matter of fact, this effect has been shown to be temporary and not effective or sufficient to break up old cultures and acquired habits. [52]. However, social interactions in the digital community are not intended to produce empowerment or autonomy for individuals, but to generate community awareness, creating a sense of belonging, interdependence and mutual support among the participants, shortening the distance between professionals and patients, resulting in greater adherence and engagement in health care and its further maintenance [65].

In the majority of the presented studies, the educational intervention strategy for individual empowerment was an objective search for effective results in changing harmful behavior in the face of conservative culture. Nevertheless, no study was found that associated social determinants in health to effective educational intervention, which could possibly bring greater coverage to intersectorality. [43,48,50,64,67].

Despite being aware of the insufficiency of individual empowerment [52,65], as well as the need to act with a broader scope, there is very little practice (practical) evidence of this in intersectorality. The 2010 Household Budget Survey, Pesquisa de Orçamento Familiar (POF), correlated the total calorie intake and income-related issues, but no inferences or even comments were present in the results, nor within discussions [51,56,59]. Race, likewise, was not even considered in the analysis of the results, yet in the Latin American context, the fact of being Caucasian was favored for having racial health-promoting factors, unlike the African who were impaired for having their health more vulnerable due to his race [64].

Lin et al. used the rigor of the evidence-based biomedical protocol to certify the benefit of physical activity on cardiometabolic health. The authors, aware of possible bias due to subjectivities, avoided any further analysis on self-referencing IPAQ [57]. At the same time, this study used the social determinants of health, with the collective focus on social support and healthy eating environments in the community. Regarding socioeconomic issues, this study establishes a correlation between a family’s per capita income (household income per capta) and physical activity practices, identifying a greater relationship between higher income and physical activities) and, lower income with sedentary lifestyle. The study did not present any data revealing statistical significance regarding the cultural issue as in the previous study, regarding racial ethnicity (white, mixed, black, Indians, Asians) [57].

Some studies grouped several elements from different fields and sectors in an attempt to act in a more intersectoral manner. This included: mental health/psychology, physical/educational activity, educational/nutritional activity; VIGITEL (Surveillance of Risk and Protective Factors for Chronic Diseases by Telephone Survey (Vigitel)), communication and safety. Very few studies on practical actions were well grounded in theoretical/conceptual aspects linked to practical applications in the fight against sedentary lifestyles and bad eating habits [43,44,46,61].

Some actions, such as “*Agita* or Shake São Paulo” had a triggering effect, an awakening, resulting in further results with global impacts. Through these initiatives, a significant part of the population abandoned a sedentary lifestyle to perform physical activities and a few of them even changed their eating habits to healthier ones, among others results observed [53,58].

### 3.3. Risk of Bias in Studies

Risk of bias analyses were carried out, verifying possible flaws in included papers to ensure the certainty of the general evidence, according to the requirements of the ROB (Risk of Bias—RoB 2—9th edition: Cochrane Bias Risk Analysis Protocol) [37]. Figure 1 presents the representation of the general analysis of 27 studies.

Figure 1 outlines the overview of the risk of bias analysis from the judgments of each dimension of risk of bias. The opinions issued are the result of the levels of “low risk”, “some concern” and “high risk”. The percentages of each grade indicate the presence of “high risk” in 22.3% of the studies; of “some concern” in 30.2% and “low risk” in 47.5%.

The ROB analysis demonstrates in percentage terms the prevalence of “low risk of bias”, as shown in Table 2 and Figure 2.

The results presented in Table 2 demonstrate, in percentages, the incidence of risk of bias levels identified in the five dimensions of analysis in the 27 studies.

These individualized results were submitted to a normality test shown in Figure 2, which graphically and statistically demonstrated a CI of 95% and a significance of p=0.69, with a mean of 79.9% and a median of 85%; 5% with low risk of bias.

The variability of the data distribution indicates “low risk” with greater homogeneity in the distribution, as well as in the order of the worksheet. “Low risk” has the lowest prevalence in randomization, with 57%, given the fact that not all studies have randomized processes. The other dimensions are above the average of 79.9%.

“Some concerns” and “high risk” occurred at average percentages of 14.18% and 5.94%.

With confidence in the body of evidence presented, the following subsection summarizes the results with the application of meta-analysis.

### 3.4. Results of Individual Studies

For the presentation of the synthesis of the results, meta-analysis was applied in accordance with the research protocol. The studies composed of parameterized data provided sufficient elements for statistical inference by pooling the analyses to create an odds ratio and a forest plot [33,71,72].

A meta-analysis of the different biomarker subgroups was performed in order to obtain more detailed outcomes, assess the effectiveness of the interventions, and achieve greater accuracy in the inference of the results [33].

An estimate of individual studies, considering the dichotomous results adjusted with Peto method, was formulated for investigating the combination or interaction of the applied studies regarding the first objective hypothesis on effectiveness [40].

Once the calculations were defined, based on the information from the findings, a general meta-analysis of the studies was performed using Review Manager software (RevMan® version 5.4.1) to verify the outcomes presented in the studies [33,72]. When entering the data, relevant adjustments were performed through calculations for parameterization. Articles that did not present data that allowed inference and adjustments were considered ceteris paribus when carrying out the meta-analysis [33,71,72].

The results shown in Figure 2 reveal the effect of the combined odds ratio by the Stuart–Kendall correlation coefficient fixed at $\mathrm{Tau}{}^{2}=1.82$. Therefore, the heterogeneity is confirmed by $I{}^{2}=100\%$, with the evaluation of the differences estimated by $\mathrm{Chi}{}^{2}=5312.60$, with gl=17 (degrees of freedom) and significance p<0.00001, confirming the symmetry, which is visually noticeable on the forest plot. With regards to the assumption of equality in the findings of the primary studies (null hypothesis), this hypothesis is refuted [71,72]. The heterogeneity is the result of several factors, among those, the variance sample size of the studies and the results of the experiments were unsuccessful due to low adherence [48,50,56,57,62,67].

There are, however, successful studies on the results of interventions, especially those by Burlandy, Ramos and Lin [45,57,61], which were effective, with an overall result of 0.60 and IC=95%, with variance from 0.32 to 1.13 for a significance of p=0.12 in the general test. The positive result of the effect of the interventions (Z=1.58), marked by the diamond, indicates the effectiveness of the interventions [71,72]. Such general effectiveness, better understood by the results of biomarkers meta-analysis, observed in Figure 3 and Figure 4 below.

The results of biomarkes meta-analysis presented in the studies were performed for better reading and assertiveness of the review analysis. Considering their relevance in determining the results, the multivariate measure of the mean of the differences between the analyzed groups was established. The heterogeneity in the general pooled test was: $\mathrm{Tau}{}^{2}=5.41$ and $I{}^{2}=92\%$.

The evaluation of differences were estimated as: $\mathrm{Chi}{}^{2}=687.43$ and gl=55, with significance p<0.00001 with symmetrical distribution (Figure 4), refuting assumptions of equality in the findings of the primary studies. The differences estimated between the subgroups were $\mathrm{Chi}{}^{2}=96.82$ and gl=6 for a significance p<0.00001, with the threshold identified by $I{}^{2}=93.8\%$.

The results presented indicate that the effect of the combined odds ratio had a weighted mean difference of −4.46 for IC=95%, with a variance of ±0.96, revealing that the interventions in the general test were effective, with a score of Z=9.06 for a significance p=0.00001. Visually, in the forest plot, the diamond in the summary of the results has the narrow appearance of a bar, given the minimum variance of −5.42 a−3.49  points in the weighted average.

The results of biomarker categories show a reduction of all items evaluated, except HDL cholesterol (mg/dL^−1^), with IC=95%, showed a minimum reduction of $-0.26\;\mathrm{mg}/{\mathrm{dL}}^{-1}$ and a variance of $\pm 4.27\;\mathrm{mg}/{\mathrm{dL}}^{-1}$; in the test for general effect, it showed a score of Z=0.12 and p=0.91.

With the diamond set on the null line in the forest plot *(*Figure 3*)* and concentrated in the top-center of the funnel plot (Figure 4), the expected decrease in high-density lipoproteins (HDLs), which are inversely proportional to the other biomarkers, did not occur. Rather, there was an increase, considering that the ideal HDL rate is greater than $40\;\mathrm{mg}/{\mathrm{dL}}^{-1}$ [74]. The results of the studies on the evaluation of HDLs were higher than expected, and therefore satisfactory.

BMI (body mass index) had a weighted average reduction of $-0.91\;\mathrm{kg}/m{}^{2},$ with a variance of $\pm 0.26\;\mathrm{Kg}/m{}^{2}$, and a Z score =2.74, for a significance of p=0.006. Systolic blood pressure showed a weighted mean reduction of −5.05 mm Hg with a variance of ±4.36 mm Hg in the test for general effect and a score of Z =2.27, for a significance of p=0.02. Diastolic blood pressure had a weighted mean reduction of −2.89 mm Hg with a variance of ±1.84 mm Hg in the test for general effect and a score of Z =3.09, for a significance of p=0.002. Total cholesterol levels had a weighted mean reduction of −18.54 mg/dL with a variance of ±4.31 mg/dL in the test for general effect and a score of Z=8.45, for a significance of p<0.00001.

Although the triglyceride rate showed a reduction, with a weighted mean of $-4.52\;\mathrm{mg}/{\mathrm{dL}}^{-1}$, the variance of $\pm 12.31\;\mathrm{mg}/{\mathrm{dL}}^{-1}$ kept the results on the null line in the forest plot *(*Figure 3*)* and gave them dominance in the funnel plot (Figure 4). Note that in the test for general effect, the score was Z=0.72  for a significance p=0.47; that is, even with the reduction, the interventions were not effective in reducing the triglyceride rate, which was above the ideal in each of the studies [75,76].

As for LDL cholesterol (low density lipoprotein), there was a reduction, with a weighted average of −4.46 mg/dL and a variance of ±0.96 mg/dL in the test for general effect, with a Z=5.74 score for a significance p<0.00001.

After presenting the results of biomarkers meta-analysis (Figure 3 and Figure 4), textual meta-analysis was conducted by Iramuteq 0.7 alpha 2 software, checking basic lexicography for word frequency, multivariate analysis of descending hierarchical classification and analysis of similarity, based on Reinert’s statistics given by the formula $P\left(r\right)\approx \frac{1}{\mathrm{rIn}\left(1.78R\right)\;}$, where “r” is the number of different words in a linguistic corpus [35,77].

By compiling the studies of the textual corpus and applying the analysis procedures in [35], a descending hierarchical classification of 120 text segments was obtained and classified on 144 (83.33%), 4784 occurrences, 1530 forms and 803 hapax (words that affect only once). Theoretical contributions were identified in the studies, making it possible to infer six classes, grouping the articles according to theoretical component, accessories and contextualization, typical characteristics and structure [35,41].

The similarity analysis, graphically represented in Figure 3, allows the visualization of the structure and its relationships with the texts and themes, and how they are distributed and connected according to their importance, demonstrating the proximities and distances, as well as the arrangement of words connected to each other, which in this case was presented in the form of a tree, with its branches [77,78].

The similarity tree depicts the structural relationships of the terms reported in the studies, with the rings (halos) indicating the groupings according to the strength of the connections of each set and how they relate between words and sets of halos, by the thickness of the branches showing the strength of incidence of the relationship. This allows the reader to identify the word “health” in the center, with its halo including core issues such as NCDs. The branches interconnect to the axes of the halos, such as “intervention”, “control”, “adherence”, “physical activity”, “obesity”, “environment”, “healthy”, “Brazil” and “study”. It is possible to observe that obesity is strongly related to prevention strategy policies, but concomitantly, distant from risk studies related to hypertension and cardiovascular problems. State programs and public actions to encourage exercise extend in an almost separate dimension, with a weak intersectoral relationship. On the other hand, interventions tend to be closer to dietary control and education. However, paradoxically, adherence is not directly related to health promotion, but to physical activities, gyms and aesthetic concerns.

### 3.5. Study Quality and Evidence Certainty

The selection and extraction process included quality assessment and sensitivity analysis, removing studies with potential bias and keeping those that have shown consistency and low risk of bias, at $79.9\%\;\left[95\%\;\mathrm{IC};\pm 21.106\%\right]$ (Figure 1, Table 2 and Figure 2). The meta-analysis of the test review included 18 studies and 148,906 participants; the odds ratio (OR) was $0.60\;\left[95\%\;\mathrm{IC};\;0.32,\;1.13\right];\;p=0.12$; heterogeneity was confirmed by $I{}^{2}=100\%$ with symmetrical distribution (Figure 2). The pooled meta-analysis with biomarkers had nine studies and seven groups of biometric markers, with 40,471 participants; the odds ratio (OR) was $-4.46\;\left[95\%\;\mathrm{IC};-5.42,-3.49\right];\;p=0.00001$. This confirms the effectiveness of interventions that reduce NCD risk rates (Figure 3 and Figure 4). The textual meta-analysis of the 27 studies, comprising more than 100 municipalities, three regions and six Brazilian states, along with two papers referring to actions in the United States and Latin America, showed $P\left(r\right)\approx 83.33\%$ (Reinert), meaning low connectivity and absence of intersectoral relationships (Figure 3). The results are consistent, making the evidence high quality, and further research is unlikely to change our confidence in the estimation of the results.

## 4. Discussion

There are challenges to overcome in developing and meeting SDG targets to ensure safe, nutritious and sufficient food (Goal 2.1), to reduce forms of malnutrition related to overweight or obesity (Goal 2.2) and to ensure adequate health development (Goal 4.2) [22]. To achieve these goals, there is a need for management and coordination of intersectoral actions.

### 4.1. Biomarkers: The Effect of Interventions

The effectiveness of NCD prevention actions was obtained by measuring biomarkers [7,9,46]. Additionally three essential components of the strategic plan to combat NCDs in Brazil were incorporated, the first two are: *“(a) monitoring risk factors; and (b) monitoring disease-specific morbidity and mortality*” [7] (p. 15).

Risk factor monitoring occurs through the observation of biochemical and inflammatory factors. The evidence makes it possible to mark the inflammatory process and the connection between metabolic syndrome and cardiovascular diseases [80]. In this sense, it includes the measurements of BMI and waist circumference, the OOCL—Overweight and Obesity Care Line, obesogenic environments and hypercholesterolemia [19,44,59]. It also includes the application of protocols such as HOMA, IR (resistant insulin) and insulin and glucose in diet [42,54,66], and evaluations of arterial hypertension through systolic and diastolic blood pressure, cholesterol, triglycerides and high and low density lipoproteins (HDL and LDL) [18,62,67,81].

Three items—variation in sample size, age, and social group —were determinants for heterogeneity, but did not alter the results. Effectiveness in reducing the rates of risk factors was proven: $-4.46\;\left[95\%\;\mathrm{IC};-5.42,-3.49\right];\;p=0.00001$.

Solutions centered on a biomedical approach were proven effective with a highly innovative technology base. On the other hand, technologies that require great expertise and high-precision medicine, imply high-cost services. Thus, these highly effective resources, which guarantee quality of life and longevity, are available only to a small group of people with privileged financial conditions, but exclude the vast population in need of the less sophisticated, but effective health services.

### 4.2. Intersectoral, Innovative and Effective Actions in the Prevention of NCDs

The third essential component of the strategic action plan to tackle NCDs in Brazil is: *“health systems responses, including management, policies, plans, infrastructure, human resources and access to essential health services, such as medicines.”* [7] (p. 15). This strategic component consists of interdisciplinary knowledge and coordinated intersectoral actions [9,82].

Some studies have displayed intersectoral actions through health sector partnerships in education, contributing to the suppression of obesogenic environments and the promotion of healthy habits such as healthy eating in the school curriculum in an innovative way [42,60]. Nonetheless, there were studies where not a single element of intersectionality could be detected [47].

Evidence pointed out to ineffective programs to combat obesity and its comorbidities, where there is an almost irreconcilable gap between primary care policy guidelines and strategies and the praxis in care facilities [45]: “*difficulties of intersectoral articulation*” [45] (p. 9), “*occurring in a punctual way around specific events—which does not leave the paper*” [45] (p. 11). It was found that “*data... are rarely used for planning local actions*” [45] (p. 11).

Interventions with some degree of multisectoral and multi-professional involvement are evident, but they seem to be divorced from intersectorality. Recognizing that obesity is conditioned by multiple factors, “*for the most part... the joint, integrated approach is rare*” [45] (p. 11). Intersectorality is still being constructed in terms of practical actions, and conceptual misconceptions between “*interdisciplinary actions*” or “*intersectoral actions*”. “*Interdisciplinarity exists above all as a practice. It translates into carrying out different types of interdisciplinary research experiences (pure and applied)”* [83] (p. 225). However, “*interdisciplinarity also concerns the cognitive activities carried out”* [83] (p. 230). *“Intersectorality is the articulation between subjects from different sectors, with different knowledge and powers in order to face complex problems”* [84] (p. 193). Interdisciplinarity is linked to the knowledge and skills involved in actions and concerns the joint articulation of sectors [85,86].

Difficulty in attributing relevance to intersectoral integration in specific actions effectively limits actions in the logic of perpetuity and results. “*The consolidation of the intersectorality of public policies began to gain value as the expected efficiency, effectiveness in the implementing sectoral policies was not observed*.” [86] (p. 1267). Intersectorality “*in the field of health, can be understood as an articulated form of work that intends to overcome the fragmentation of knowledge and social structures to produce more significant effects on the health of the population*” [84] (p. 193).

The studies’ limitations are evidenced in the lack of intersectoral elements, as well as in the absence of relationships between health practices and social practices. Elements such as socioeconomic status and literacy level, which apparently are close to social practice, are subjective and sensitive elements that interfere in the field of objectivity and concreteness, and can even be contradictory and opposing elements in a broader analysis of the complex issue of obesity. Unfortunately, the vast majority of interventions include social determinants of health variables in a simplistic way as pragmatic instrumentation, far from a real understanding of how the use of intersectorality could effectively interfere to change the this endemic condition.

Future studies should be carried out, not necessarily systematic reviews, but regional and national searches for municipalities that have taken actions on the issue of non-communicable chronic diseases that may be intersectorial, linked and carried out within multiple sectors of society such as planning, education, legislation, agriculture, environment, tourism, etc. These certainly deserve attention in the analysis of effectiveness and innovation in order to verify the possibility of replicating them in other municipalities.

It is recommended for future studies the conducting of an integrative review of the data to be obtained, not from scientific research data repository, but from Ministry of Health (SUS) repositories, such as National Council of Health Municipal Secretaries (Conselho Nacional dos Secretários Municipais de Saúde – CONASEMS) and National Council of Health Secretaries (Conselho Nacional de Secretários de Saúde - CONASS), with the following contents with the retrieved data on: the successful actions in combating NCDs in municipalities across the country, such as management and planning, health education, social media and community, health promotion, direct or indirect interventions on obesity and comorbidities. From the perspective of intersectoriality, in a more in-depth analysis, with the objective of finding, perhaps not new social, sociological, anthropological, economic, political elements, etc. but to understand how these combined elements would affect in the management and monitoring of NCDs, to become either promoters or detractors of health.

## 5. Conclusions

The present study aimed to identify innovative and effective intersectoral actions for the prevention of NCDs in Brazilian municipalities. From the hypotheses that guided the research, it can be concluded: (1) As for the effectiveness of interventions, $H_{0}$ is accepted, because interventions for the prevention and treatment of obesity, diabetes and hypertension were effective in reducing risk factors by $-4.46\;\left[95\%\;\mathrm{IC};-5.42,-3.49\right];\;p=0.00001$, which represents a high impact on the weighted average of risk factors. (2) Regarding the presence of innovative and intersectoral elements, $H_{0}$ is rejected, therefore assuming $H_{1}$—absence of innovative and intersectoral elements in actions to combat and prevent NCDs, with $P\left(r\right)\approx 83.33\%$ (Reinert), considering the low connectivity between the ‘halos’ of actions and the absence of intersectoral relationships. (3) Replicability, $H_{0}$, is rejected, given the fact that it depends on the second hypothesis; therefore $H_{1}\;\mathrm{is}\;\mathrm{assumed}$ to be effective, innovative and intersectoral actions are not replicable, even with adjustments.

The meta-analysis corroborates the finding of unresolved problems in the consolidation and continuity of existing programs. The studies revealed the effectiveness of the interventions, unlike the intersectoral actions, which can be seen with P(r) ≈ 83.33% (Reinert), considering the low connectivity between the ‘halos’ of actions and the absence of the intersectoral relationships.

## Data Availability

The systematic review follows the Preferred Reporting Items for Systematic Reviews and Meta-Analysis (PRISMA 2020) guidelines and is registered in the PROSPERO database with the ID: CRD42022319835, with research protocol and data management plan in the DMPTool database with DOI: 10.48321/D1K31N.

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
