# Peer review of "Intersectoral Actions for the Promotion and Prevention of Obesity, Diabetes and Hypertension in Brazilian Cities: A Systematic Review and Meta-Analysis"

_ijerph, 2022, doi:10.3390/ijerph192013059_

Round 1
Reviewer 1 Report
1. In the introduction, references 1-6 are out of date and it is recommended to update the latest evidence to elucidate.
2. All the Research Protocol and Data Management Plan via the link: https://dmphub.cdlib.org/dmps/doi:10.48321/D1K31N were presented in non-English language, so it is difficult for me to understand them. It is suggested to convert them into English for better interpretation.
3. As a systematic review and meta-analysis, it is necessary to register in advance for the implementation protocol on the Cochrane or PROSPERO to ensure that this work is practicable, free from reporting bias, etc. Have you registered yet? If so, provide your registration number.
4. The details of inclusion and exclusion criteria should be described in the section of method.
5. All the processes of data selecting and quality assessment were conducted by one person? How does this avoid the subjective judgement of one person? Please clarify.
6. The language of this review needs to be polished by a native English speaker.
Author Response
Dear Reviewer,
After extensive work of adjustments and adaptations as directed, the updated version of the work is attached.
As per your observations:
1. In the introduction, references 1-6 have also been updated with more recent evidence.
2. We regret that you did not have access to the English version of the Data Management Plan on the DMPTools Platform. We provide a copy in English which is attached.
3. Yes, the protocol is registered at PROSPERO, and it is duly referenced in section 5.1 in the Data Availability Statement item. We also include referencing in section 2.2. To facilitate, we provide a copy that follows attached.
4. The inclusion and exclusion criteria are described in the method section in a summarized way, even though it consists of 870 words. All the detailed content can be accessed in the Research Data Management Plan registered in both PROSPERO and DMPTools, duly referenced in sections 2.2 and 5.1.
5. All data selection and quality assessment processes were conducted by peers, discussed by all authors and with some collaborators mentioned in the acknowledgments. Thus, avoiding subjectivity and/or the interpretation and understanding bias of a single researcher. Searching in the plurality of understanding the consensus for the accomplishment of the research report.
6. Following your guidance, language adaptation of the Text was made by Stephen Kunihiro, Ariane Nishimura Kunihiro (translator), and Hannes Fisher (native in English).
We appreciate your guidance and make ourselves available for any information.

Reviewer 2 Report
Thank you for the opportunity to review this article.
The aim of this study is to assess the effectiveness of biomedical interventions and to identify intersectoral innovative and effective actions for the prevention of obesity, hypertension and diabetes in Brasil. The conclusions of this paper are interesting, important for the Public Health, and some of these can be applicable to other countries.
General comments
1.The results are clearly present, revealing the importance of prevention strategies in NCDs, with the absence of intersectoral relationship. The majority of the 27 studies used, had a relatively short follow-up period and some studies had a small number of patients(<100).
2. Some comments should be added about the effectiveness of treatments/ actions ( partially/total satisfactory I2, I3) in the age group under 19 years old (E1, E2, E3) from the Table 2.
3. Be more specific about the impact of the Socioeconomic indicator (D2), Educational level indicator (D3) and the Cultural issues(D4) on the results.
Author Response
Dear Reviewer,
Thank you for the guidance and observations in our study.
1. We have unified the results of the first two tables into a single one in order to provide a better elucidation of the results, especially regarding the size, time and application of the experiments.
2. We have improved the comments on the analyzes that continue to elucidate (partial/total satisfactory I 2, I 3 ), age group issues and the application of intersectoriality in actions. We have carried out a new version of the writing of section 3.2 Characteristics of the Studies.
3. In the new wording, we seek to better clarify the analysis of the Indicators of the entire Table 1, which replaces the previous Tables 1 and 2.
We appreciate your guidance and make ourselves available for any information.
